# Three-Questions-Method for Coping with the Emotional Burden of Nurses and Nursing Students during COVID-19

**DOI:** 10.3390/ijerph19116538

**Published:** 2022-05-27

**Authors:** Cindy M. A. de Bot, Annemarie J. B. M. de Vos

**Affiliations:** 1School of Health Studies, Avans University of Applied Sciences, 4818 CR Breda, The Netherlands; jbm.devos@etz.nl; 2Amphia Hospital, 4818 CK Breda, The Netherlands; 3Elisabeth-TweeSteden Hospital, 5022 GC Tilburg, The Netherlands; 4School of Health Studies, Fontys University of Applied Sciences, 5631 BN Eindhoven, The Netherlands

**Keywords:** nurses, nursing students, emotional burden, COVID-19

## Abstract

Background: Nurses and nursing students experienced an emotional burden while working during the COVID-19 outbreak. During the COVID-19 outbreak three questions for nurses working under these extreme circumstances were formulated: 1. What today’s events do you remember? 2. How do you feel (physically and mentally)? 3. Do you have enough support? The purpose of this study was to obtain insight into whether nurses and nursing students perceive that the use of the three-questions-method contributes to effective coping with the emotional burden during the COVID-19 outbreak. Methods: Focus group interviews were held with hospital nurses (n = 11) and nursing students with internships in mental health care (n = 2), hospital (n = 9), and homecare/nursing home care (n = 3) in September 2020 followed by twenty semi-structured interviews one year later. Results: Almost all nurses and nursing students named factors that contributed to the emotional burden: fear, powerlessness, frustration, lack of knowledge about COVID-19, and pressure to pass the internship. Participants indicated that using the three-questions-method can help to effectively cope with the emotional burden during and after the COVID-19 outbreak. Conclusions: Using the three-questions-method offers added value in coping with emotional burden and can be used in education as well as in practice.

## 1. Introduction

The coronavirus crisis in the Netherlands was part of the global coronavirus crisis, which was caused by the outbreak of the infectious disease COVID-19 in 2019–2020. At the beginning of 2020, SARS-CoV-2—the coronavirus that causes COVID-19—was also detected in the Netherlands. Although the first infection was officially detected on 27 February 2020, a number of Dutch people had already been infected.

The outbreak led to measures that were unprecedented in the Netherlands, whereby social traffic was largely halted. Schools, universities, libraries, cafes, hairdressers, museums, cinemas, and restaurants had to close their doors by government directive. High school exams, sports matches, and many other events were cancelled. Dutch people were advised to keep a distance of 1.5 m from each other and to refrain from traveling abroad. In addition, people were asked to stay indoors as much as possible and to work from home. Foreigners were instructed to only travel to the Netherlands if strictly necessary [1].

Working in healthcare during the COVID-19 outbreak in 2020 caused a significant emotional burden for nurses and nursing students (hereafter referred to as nurses). The enormous influx of patients during the first wave in 2020, the long working days, and the erratic course of the disease caused feelings of powerlessness, stress, and fear in nurses [2,3,4,5]. During and after the COVID-19 outbreak, support for nurses and nursing students should be given in all areas of care [5].

A reservoir of delayed care with patients sicker than before awaited after the first major COVID-19 wave. As a result, the amount of work for nurses doubled, as they delivered not only usual care, but also the deferred and high complexity care to the remaining COVID-19 patients [6]. A poll of 3000 nurses in August 2020 demonstrated that a large proportion of Dutch nurses still experienced the effects of the 2020 spring COVID-19 wave. Some were sick at home with burnout or recovering from COVID-19 [7]. Half of the respondents were not in shape to face another COVID-19 wave. Meanwhile, nurses faced the third wave COVID-19 patients, usual care was scaled down, and staff shortages (absenteeism and shortage of trained staff) were the main bottlenecks [8]. Even in 2022, the nursing workforce is still working in an ongoing crisis situation, due to the added stress of COVID-19 and deferred care.

Collard and Vermeulen formulated a set of three questions for nurses working in the extreme conditions: 1. What today’s events do you remember? 2. How do you feel (physically and mentally)? 3. Do you have enough support [9]? The three-questions-method is meant to be used as an aid to raise the right issues in a conversation in order to cope with the emotional burden. Coping is defined as the thoughts and actions that individuals use to deal with stressful events [10,11]. The questions are formulated in a structured manner, starting with a concrete question, which is followed by an internally focused question about emotions. The third question refocuses on the exterior. It is anticipated that just asking these questions suffices to start a conversation. The objective of this study was to obtain insight into whether nurses and nursing students perceive that the use of the three-questions-method contributes to effectively coping with the emotional burden during and after the COVID-19 outbreak.

## 2. Materials and Methods

### 2.1. Study Design

This qualitative study was conducted in two phases over two consecutive years (Appendix A). In phase 1, focus groups were held to obtain insight into the experienced emotional burden among nurses while working in healthcare during the COVID-19 outbreak in 2020. In addition, the study explored the potential benefits of using the three-questions-method in coping with the emotional burden among nurses during COVID-19.

Phase 2 investigated the extent to which the three-questions-method was used in the Netherlands and explored the nurses’ perceptions with regard to the effectiveness of the method. Qualitative studies offer a rich description of a specific phenomenon, particularly when not much is known about it (i.e., the outbreak of a new disease) [12]. Qualitative designs involve broad research questions that capture the various dimensions of a new situation or event, enabling researchers to present a holistic and rich description [13].

### 2.2. Study Population and Setting

In phase 1 of the study, we purposively invited nurses and (year 2–4) nursing students who were involved in nursing care during the COVID-19 outbreak in 2020. To maximise the generalizability of the findings, we selected a heterogeneous group working in various departments and health care settings. The study was conducted in a large teaching hospital and at the University of Applied Sciences in the Netherlands in September 2020.

In phase 2 (in 2021), we invited all Dutch nurses to participate by using a web link, which was disseminated via institutional websites and Linkedin. The link provided the description of the study followed by open questions to guide the participants’ narrations, including an invitation for an online in-depth interview. All participants provided written informed consent by email to participate in the study.

### 2.3. Data Collection

Data were collected face to face or using video links due to pandemic restrictions in 2020 and 2021. An interview guide for both the focus groups (Appendix A phase 1) and interviews (Appendix A phase 2) was developed and utilized to lead the conversations to the study areas by an independent unbiased moderator. The moderator served as discussion leader and was responsible for guiding the participants through the discussion. In addition, she ensured that all participants joined in the discussion in order to balance participation. The participants were encouraged to express their views and opinions freely. The average time for an interview was 45 min, ranging from 30 to 60 min. The online interviews (phase 2) were audiotaped via Microsoft Teams with the participants’ permission and transcribed verbatim and anonymously.

### 2.4. Data Analysis

To identify, analyse, and report patterns in the data, trained and experienced researchers CMAdB and AJBMdV performed an inductive thematic analysis in phases 1 and 2 [14]. They used an essentialist analysis method to ensure reporting the participants’ reality. Furthermore, they used a rich thematic description to maximize insight in all important concepts. The development of the coding structure was an iterative process [14]. First, CMAdB and AJBMdV read and coded the transcripts using the principle of open coding of data. Next, they discussed the results and determined the final codes and code groups. In regular meetings, CMAdB and AJBMdV discussed discrepancies to reach consensus about the refinement of codes and code groups. Finally, they merged the identified code groups into themes based on similarities. To maximize reliability and integrity of the results they performed a member check, which enhanced the trustworthiness of the results [15]. In phase 2, the data were also systematically reviewed by CMAdB and AJBMdV to ensure that a name, definition, and exhaustive set of data were identified to support each category. To ensure dependability, all research steps, including data collection, data analysis, and manuscript preparation, were well-documented. Continuous reflections, in particular potential preconceptions, were discussed between the first and second author. To illustrate the original data and enhance the description of the categories, the results section presents relevant excerpts from the interviews.

### 2.5. Ethical Considerations

The study was designed, planned, and performed according to Dutch law, stating that ethical approval is not needed when healthcare professionals are asked to participate in research about work-related questions. All participants were informed orally and in writing about the study aim, the fact that participation was voluntary, and that they could withdraw at any time without negative consequences. Confidentiality and anonymity were ensured by removing any identifying information before processing the data.

## 3. Results

In phase 1, 20 hospital nurses and 14 nursing students undertaking internships in mental health care (n = 2), hospital (n = 9), and homecare/nursing home care (n = 3) participated in the focus groups. We aimed to obtain insight into the experienced emotional burden and explore the potential benefits of using the three-questions-method.

Phase 1: Experienced Emotional Burden.

Participants named four factors that contributed to the emotional burden: (1) fear and powerlessness, (2) frustration, (3) pressure due to curricular requirements, and (4) physical burden.

### 3.1. Fear and Powerlessness

Participants were anxious about becoming ill themselves, but also feared infecting patients or family members. In addition, they stated that at times they were not able to do anything for patients, which led to feelings of helplessness. The death of a large number of patients and clients also contributed to these feelings.


*Because then you’d have a day and then, for example, three of the patients you cared for that day died on your shift. And then you start thinking to yourself, like: what did I forget? Or what haven’t I done?*
[#F1]

### 3.2. Frustration

The feeling of frustration was caused by the attitude of the outside world and media reports, according to the participants. As a result, they stopped following news reports and current affairs programs to protect themselves. In addition, the lack of knowledge and experience about the new syndrome led to frustration in practicing the profession.


*It also made me angry. But also the aggression, they write about a lot in the newspaper now. Yes, daily you get a big mouth and threats to your head from visitors because you address them about the COVID policy.*
[#F2]

### 3.3. Pressure due to Curricular Requirements

Nursing students mentioned that they felt pressure from complying with the assignments from the university. In addition to COVID-19 patient care, they were responsible for their learning and development, such as completing the internship and conducting hands-on nursing research.


*I still have an internship, I still have to do certain things, I feel the pressure from university.*
[#F4]

### 3.4. Physical Burden

Factors related to the physical burden were also mentioned. Almost all participants reported being overtired. In addition, they explained that they had no time to drink or eat during the work shift. Moreover, they indicated that, due to the long working days, they were physically exhausted after a shift.


*I’d come home at, I don’t know, six o’clock at night or something after a shift, and I’d lie down in my bed and I’d fall asleep until the next morning. And then I hadn’t even eaten, because I was exhausted and I just couldn’t take it anymore.*
[#F2]

### 3.5. Support

Peer support or support from colleagues was mentioned as a way to reduce the emotional burden, but also support from family and friends was seen as helpful. However, participants also stated that family members often did not know what was going on in the hospital.


*Well, we often talked about it with colleagues. How it was for us as nurses. And we still have quite a few conversations about it.*
[#F3]

Although nursing students felt supported by teachers and fellow students, they explained that online education limited the interaction with lecturers and students.


*That you actually see each other and it’s different anyway, because online you’re going to take into account that someone else might want to say something. You’re going to hold back because you’re afraid you’re going to talk through someone and then you end up saying nothing.*
[#F4]

### 3.6. Solidarity and Resiliance

Nurses but also other healthcare professionals reacted quickly to the pandemic and readied themselves to collaborate in patient care. Nurses’ adaptability and sense of belonging during work were perceived as positive.


*We all stood for the same thing and everyone for the same thing. The cleaners or the people from the catering: everyone for the same goal. And there was a great togetherness. I liked that.*
[#F3]

Phase 1: Potential Benefits of Using Three-Questions-Method.

During the focus group interviews in phase 1, participants indicated that the three-questions-method could be used proactively and informally within education, for example during intervision meetings, work supervision, or student career counseling. They also suggested that—in nursing practice—the three-questions-method could be used during the coffee or lunch break, handover moments, or evaluation sessions. According to the focus group participants, both supervisors and colleagues could ask the questions back and forth.


*“I think that if you ask people those three questions that you do open up a conversation more quickly. First of all, that people start to express themselves more easily than with the question “how are you?”*
[#F2]

In phase 2, 15 Dutch nurses and 5 nursing students participated in the interview sessions, in which we investigated the extent to which the three-questions-method was used and the perception of the effectiveness of using the method. A year after the introduction, 6 of the 20 participants used the three-questions-method.

Phase 2: Perceived Effectiveness of Using Three-Questions-Method.

All respondents (n = 20) indicated that the COVID-19 period had impacted them, both physically and emotionally. In particular, the emotional burden took a substantial toll on the respondents. They explained that there was not enough time or space to express their emotions sufficiently. A large proportion of the respondents indicated the need for stability and a safe working environment in which they can talk to trustworthy colleagues.

### 3.7. Experience with Using the Three-Questions-Method

Six participants mentioned having a positive experience with the use of the three-questions-method. According to them, the method ensures that experiences are made discussable, as team members have the opportunity to express their feelings. According to some respondents, feelings of stress and insecurity also diminish by using the three-questions-method. Three users were very positive about the three-questions-method because of its easy use. According to them, the three-questions-method is simple and straightforward to use, which gives everyone the opportunity for a short evaluation.

Three respondents mentioned gaining more insight into their colleagues’ experiences, which could provide a handle on how to deal with problems in the team. In addition, they mentioned that the issues raised while using the method could create a sense of trust and security in a team in the long run.


*It’s actually a bit of insight into each other’s heads. That’s how I actually see it. And where you can help each other and where you can support each other as a team.*
[#I19]

### 3.8. Facilitating and Hindering Factors

The respondents pointed out that both facilitating and hindering factors influence the appropriate use of the three-questions-method.

Working with familiar colleagues was perceived to be a facilitating factor when using the three-questions-methods. Respondents explained that sharing emotions with familiar colleagues was easier in comparison with new colleagues. Using the three-questions-method in smaller groups has a stimulating effect, because a small group encourages sharing experiences, according to three respondents. However, some respondents indicated to prefer one-on-one conversations with colleagues, due to feelings of unsafety within a group.


*Well, if you are not much of a talker, it is difficult.*
[#I9]

Trusting colleagues and feeling safe within the group facilitate the use of the three-questions-method, according to six respondents. In addition, they explained that the method can be used in a low-key way, which was considered to be pleasant.


*...that relationship of trust is essential and if the safety in the group is not there, you are not going to share in that.*
[#I11]

Phase 2: Use of Three-Questions-Method in Healthcare Organisations.

### 3.9. Use of the Three-Questions-Method (n = 6)

Two respondents indicated that they used the three-questions-method in multidisciplinary groups with—for example—a social worker and a pastoral caregiver.

Most respondents indicated that they use the three-questions-method at three fixed moments during the shift: at the start, halfway, and at the end of the shift. One respondent mentioned that the three-questions-method was not used structurally, but approximately once a month. However, the respondent suggested that this was not enough to be effective. In addition, one respondent indicated that the method was used less in a team with many new colleagues, because the feeling of trust was expected to be lower in such a team.

Three respondents indicated that the three-questions-method was mainly used in groups, but that it was also possible to use it with one or two others. Two respondents indicated that the effect of the three-questions-method depends on the context. They indicated that it was important to use the three-questions-method more often during difficult periods, such as the COVID-19 pandemic. Particularly due its simplicity, the method offers an easy-to-use tool to cope with the emotional burden, according to six respondents.

The interviews revealed that several respondents would also use the method after the COVID-19 outbreak and in multidisciplinary settings and meetings.


*And so I think it’s not just for nurses but actually for many more health care professionals.*
[#I11]

## 4. Discussion

This study demonstrated that nurses experience an emotional burden due to their work during the COVID-19 outbreak. This manifests itself in powerlessness and frustration, partly caused by the lack of knowledge and experience about the new disease and its clinical manifestation. The pressure of successfully completing the internship increases the emotional burden for nursing students substantially. Fatigue and physical complaints also contribute to the physical burden. These findings are in line with nurses’ experiences during previous epidemics of SARS, MERS-CoV, and Ebola, during which nurses developed physical and mental disorders, such as loneliness, anxiety, fear, fatigue, and sleep disorders [16,17,18].

Peer support is an effective intervention in reducing work stress during intense situations [19,20,21,22]. The study of Agarwal and colleagues, for example, shows that peer support in the workplace may improve employees’ well-being and relationships between employees [21]. This study demonstrates that the three-questions-method is an easy to use and effective tool for coping with the emotional burden among nurses. Both users and non-users considered the three-questions-method to be non-committal and informal, which promotes its use. The majority of the sample did not use the three-questions-method, because they were not informed by their management about the method, nor did they keep up with the nursing journals (due to the experienced emotional burden in 2020–2021). Nevertheless, they anticipate that the low-key and informal use is conducive to gaining more insight into the feelings of colleagues, enabling them to support colleagues more effectively. This small-scale study emphasizes the need to implement emotional support tools for nurses in daily clinical practice to facilitate emotional de-escalation during and after stressful events.

Nursing students collaborated with nurses to deliver patient care during the COVID-19 outbreak. As a result of university closure and the switch to online education, they experienced a pause in their education. Both nurses and nursing students had to cope with mental and emotional issues, including stress, anxiety, and fear [23,24]. In our study, nursing students had to cope with stress and anxiety, but felt supported by lecturers or fellow students, despite the online education being a hindering factor in expressing concerns to lecturers [25,26]. Therefore, it is important that nursing students remain supported by academic and training programs. Moreover, universities should not only address student mental health, but also implement strategies that promote students’ understanding of crisis management, self-care, and coping skills [25].

## 5. Strengths and Limitations

This research presents several strengths. The researchers conducted the interviews via online video calls, as they could not visit the healthcare organizations and universities, due to the implemented safety measures during the COVID-19 outbreak. This method allowed for both verbal and nonverbal data to be acquired. The interviewers were trained in interviewing skills. Due to the heterogeneity of the sample (multiple health care organizations, both nurses and nursing students), the results demonstrated a broad spectrum of experiences with the three-questions-method.

The study also had some limitations. Precise and generalizable results could not be obtained, because of the qualitative design and the restricted sample size. In addition, the study was conducted in a short period of time. Longitudinal research would offer a valuable avenue for future exploration.

## 6. Conclusions

The results show that both nurses and nursing students experienced an emotional burden from their work during the COVID-19 outbreak. This was caused by a combination of factors, including fear, helplessness, and frustration. For nursing students, the perceived pressure of obtaining their internship also played a major role. Fatigue, physical complaints, and poor self-care also contributed to the emotional burden.

It is important that healthcare and educational organizations pay attention to the emotional and physical health of nurses and nursing students while working in crisis situations, such as the COVID-19 outbreak. The three-questions-method provides nurses and nursing students an easy-to-use and effective tool to cope with the emotional burden during work. If we want to retain current and future nurses, it is essential that we support them in staying emotionally and physically healthy, now and in the future.

## Data Availability

The datasets generated and/or analyzed during the current study are not publicly available due to the nature of the data and the relatively low number of participants but are available from the corresponding author on reasonable request.

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
