# Peer review of "Three-Questions-Method for Coping with the Emotional Burden of Nurses and Nursing Students during COVID-19"

_ijerph, 2022, doi:10.3390/ijerph19116538_

Round 1

Reviewer 1 Report

Dear Authors,

-The study uses a three-questions-method for coping with interviewing the emotional burden of nurses and nursing students during COVID-19. During CoVID-19, it's a chanllange work. However, some issues suggest addressing more. Introduction section, Lack the emotional burden, three-questions-method, and effectively coping related references and definition.

-please clarify the study’s actual research questions. The research questions or objectives are unclear.

-“Data collection,  Data were collected face to face or using video links owing to pandemic restrictions in  2020 and 2021. An interview guide for both the focus groups (Figure 1 phase 1) and interviews (Figure 2 phase 2) was developed and utilized to lead the conversations to the study areas by an independent unbiased moderator” the part is unclear, suggests addressing more.

-“The online interviews were audiotaped with the participants’ permission and transcribed verbatim and anonymously.” The study on how to do online interviews is Also unclear.

-Up to unclear coding book, coding themey, analysis method, and related references support, so the result complete concerns, in addition, the part “inductive thematic analysis.”Suggests addressing more on how to do the analysis and coding.

-Furthermore, lack of strength references support “…emotional burden: 1) fear and powerlessness, 125 2) frustration, 3) pressure due to curricular requirements, and 4) physical burden. …“So the analysis needs to be addressed.

- The discussion and Conclusion section may need to reword up to the unclear interview analysis method.

Thank you for your efforts.

Reviewer 2 Report

Thank you for submitting your manuscript to the International Journal of Environmental Research and Public Health.

The topic of the research is interesting and can be related to the promotion of the quality of nursing care. However, after reading the manuscript, I consider that the authors should review some points:

  • General:

IJERPH magazine has an international impact and this means that readers are from different countries with different professional backgrounds and different experiences and impacts with the COVID19 pandemic. I suggest that authors include in the introduction a description of the common and distinctive characteristics of their country with respect to other countries. This context can help readers to understand the impact or to use your study in an international analysis.

  • Methodology:

The methodology shows that there are different objectives: “to obtain insight into the experienced emotional burden”, “the study explored the potential benefits of using the three-questions-method in coping with the emotional burden among nurses during COVID-19”, “The Phase 2 investigated to what extent the three-question method was used” or “explored the nurses' perceptions with regard to the effectiveness of the method”. All issues are related but different. What is the general objective of the study? How do these questions help to answer the general objective?

In this sense, I suggest the authors accompany the manuscript with an image or flowchart that quickly clarifies for any reader the phases, the objective of each one, the difference in time and participants.

    • Participants:

The sample of participants is scarce or moderate if they have not been intentionally selected for providing a specific profile within the groups of nurses and students. If the selection is intentional, the criteria applied must be provided.

In addition, I suggest that you include in the manuscript a description of the group that includes those aspects of interest for this study (for example, did they have previous experience in epidemics or critical situations? Had they received training to improve coping?)

    • Study design:

The Study design describes the use of three questions that are detailed in the Supplementary Materials section with many more questions. In addition, it is stated that CMAdB and AJBMdV carried out the content analysis and coding work, but their profile or experience as suitable researchers for this task is not described. This methodology can be accepted when they have extensive training and experience in this type of analysis.

On the other hand, in some sentences “we” appears as if the authors were part of the analysis and coding process. Can you clarify this part of the methodology to improve the clarity of the manuscript? This would allow others to assess the quality of the study.

The methodology must allow other researchers to replicate their study, so this section must improve clarity and detail in the criteria used for the groups and the analysis of the interviews.

  • Results:

The results show an interesting situation that makes visible the impact on nursing professionals and students, however, the results can be increased and detailed. One way to improve clarity is to use the objectives proposed in each phase to summarize the results, establishing as titles Phase 1 or Phase 2, the question it answers and, finally, the sections considered by the authors. This layout, along with a picture on the methodology, will help any reader understand your interesting study.

  • I suggest including a description or answer to all the questions that have been asked or, if it is not considered appropriate to detail each item, the authors should make a global description of the result and highlight the most important.
  • The textual fragments of the results are interesting and are considered adequate to represent each of the themes extracted.
  • In phase 2 the number of participants is smaller and it seems that they are different from the first group. I request to clarify if all the participants are different from the first group and describe this sample (eg age, experience, place of work...)
  • In phase 2 the use of the methodology of the three questions is analyzed, what is the established criterion to consider that it is used? Once, twice, at least once a month, daily? In addition, was it ensured that the participants who claimed to use the methodology had done so and had asked all the questions?
  • Discussion
  • The theme is novel and similar articles may not be found in relation to the pandemic period, however, this methodology has been used in other contexts or situations. Can these studies be provided to discuss the results obtained.

I encourage authors to review the issues raised and resubmit the manuscript.

Thank you very much.

Best regards

Reviewer 3 Report

The research topic is interesting and the choice of the qualitative method is appropriate to investigate the question in all its facets. Nevertheless, this study deserves further work in terms of the structuring of the article as well as in the results section. A more in-depth analysis would bring robustness to this study and would make it possible to propose concrete avenues for the support of nurses in a period of crisis such as that of the covid 19 pandemic.

  • Introduction section:
    We fail to understand what the primary objective of this study is and what (if any?) secondary objectives are.
    Indeed, we have difficulty understanding if this is an interventional study aimed at using the 3 questions method to assess its effectiveness (its impact) with the target audience? If so, who are the interveners and what is their training/qualification for this intervention?
    If not, would it be a SHS study aiming to question the representations, perceptions and experiences of the nurses with regard to covid 19, using the 3 questions method in the focus groups and individual semi-directive interviews in order to optimize the data collection?
  • Material and Methods : The use of a COREQ (equator network) type guideline is expected to guarantee the rigor, transparency and quality of the reporting of the qualitative analysis.
  • Results section : And what about the 14 nurses who do not use the 3 questions method? What are the barriers in current nursing practice? It would be interesting to understand why less than half used this method over a 12-month period.
    Thus, it would be interesting to go back to the thematic content analysis and refine it to understand in detail why a majority of the sample did not use the method on a long-term basis. Discussing these results with the literature would be very enriching for the scientific community.
  • Discussion section : Line 281 to line 286 : It seems that this does not appear in the results section for non-users in particular. Or it is not clearly explained, which causes confusion.
  • Strengths and limitations : Line 305 to line 306 : Very surprising! A sample size of 20 is quite respectable for a robust qualitative study (cf saldana). The sample size is not a problem for robust qualitative results. The problem here is that only a small part of the sample (n=6) uses the 3 questions method and finds it efficient. n=6 cannot be representative since it represents less than half of the n.
  • Conclusions : The effectiveness of the use of the method was not proven as only n=6 out of a total sample of n=20 used it. The conclusion is not in line with the results. 

Round 2

Reviewer 1 Report

I  have no other suggestions, thank you.

Author Response

Thank you for your time to review our manuscript

Reviewer 2 Report

Dear authors

Thank you very much for reviewing your manuscript. The quality of the structure and writing has improved. I consider that this version is sufficient to consider its publication.
However, I ask you to review the bibliography section. Bibliographical references do not have the same style. For example, in the latest version bibliographic references 1, 6, 7, 8, 9, 10, 11 and 25 present the year in parentheses and the rest after the title. Please review these details.

Thank you very much

Best regards

Author Response

We have revised your comments in the references.

Thank you for your time to review our manuscript

Reviewer 3 Report

Thanks to the authors for the revised version. The work done contributes to improve the manuscript.

Nevertheless, the changes made do not seem sufficient for publication. 
The discussion section needs to be improved and enriched because it is important to shed light on the fact that the majority of the sample does not use this method: what are the obstacles? Is it linked to the system? Is it related to professional identity? We do not know if the non-users are more nurses or students? Does it make a difference in the behaviour toward the use of this method? It seems necessary to refine the qualitative analysis in this sense. 
We understand that the non-users are not opposed to the use of the method, but they do not use it. Why not? What elements of understanding can you bring? What would be necessary for this method to find its place and to be used in current nursing practice to allow a better management of emotions? What recommendations can you make?

You cannot conclude about the 'effectiveness' of this method with less than half of a sample using the method. This does not mean that the method is not viable or effective. It would be very interesting to provide some insight into why this method has not found its way into routine practice.

Thanking you for any additional work you can provide on this manuscript.

Author Response

We have revised the discussion paragraph.

Thank you for your time to review our manuscript